# Organic Matter Retrieval in Black Soil Based on Oblique Extremum Signatures

**Mingyue Zhang** [1,2] , **Maozhi Wang** [1,2,\*] **, Daming Wang** [3] **, Shangkun Wang** [1,2] **and Wenxi Xu** [1,2]

1   Geomathematics Key Laboratory of Sichuan, Chengdu University of Technology, Chengdu 610059, China; zhangmingyue@stu.cdut.edu.cn (M.Z.); wshk@cdut.edu.cn (S.W.); xuwenxi@cdut.edu.cn (W.X.)
2   College of Mathematics and Physics, Chengdu University of Technology, Chengdu 610059, China
3   Tianjin Center of China Geological Survey, Tianjin 300170, China; wangdaming@mail.cgs.gov.cn
\*   Correspondence: wangmz@cdut.edu.cn

**Abstract:** How to extract the indicative signatures from the spectral data is an important issue for further retrieval based on remote sensing technique. This study provides new insight into extracting indicative signatures by identifying oblique extremum points, rather than local extremum points traditionally known as absorption points. A case study on retrieving soil organic matter (SOM) contents from the black soil region in Northeast China using spectral data revealed that the oblique extremum method can effectively identify weak absorption signatures hidden in the spectral data. Moreover, the comparison of retrieval outcomes using various indicative signature extraction methods reveals that the oblique extremum method outperforms the correlation analysis and traditional extremum methods. The experimental findings demonstrate that the radial basis function (RBF) neural network retrieval model exposes the nonlinear relationship between reflectance (or reflectance transformation results) and the SOM contents. Additionally, an improved oblique extremum method based on the second-order derivative is provided. Overall, this research presents a novel perspective on indicative signature extraction, which could potentially offer better retrieval performance than traditional methods.

**Keywords:** soil organic matter; indicative signatures; oblique extremum; radial basis function neural network

## 1. Introduction

Black soil plays a vital role in maintaining food security. Unfortunately, the quality of black soil has deteriorated due to improper human use [1,2]. Soil organic matter (SOM) is a crucial indicator for soil fertility assessment and serves as the primary source of various nutrients necessary for plant growth [3]. Furthermore, SOM can enhance soil resilience and fertility retention, improve soil physicochemical properties, and help fix soil heavy metals and organic pollutants [4]. Consequently, accurate and timely measuring of SOM content is crucial to monitor land quality. However, traditional methods requiring field samples and expensive laboratory analyses [5,6] are impractical for large-scale land-quality monitoring. Therefore, a quantitative retrieval method based on spectral data is an efficient and viable solution to this problem.

Soil composition monitoring through spectral data analysis typically involves three major processes: spectral data analysis, indicative signature extraction, and retrieval modeling. Here, spectral data analysis ordinarily involves spectrum denoising [6,7], spectral transformation [6,8,9], standard normal variate (SNV) [7], multiplicative scatter correction (MSC) [10], and continuum removal (CR) [11–13]. As to the indicative signature extraction, correlation analysis is the most popular method [5,14]. Some other methods, such as genetic algorithm (GA) [15,16], successive projections algorithm (SPA) [8,17,18], and competitive adaptive reweighted sampling (CARS) [15], are also employed. Further, the quantitative

retrieval models for soil components with soil spectra are categorized into statistical analysis and machine learning models, including multiple linear regression (MLR) [9], multiple stepwise linear regression (SMLR) [8,19], principal component regression (PCR) [20], partial least squares regression (PLSR) [7–9,21], back propagation (BP) neural network [14], extreme learning machine (ELM) [22], random forest (RF) [9,23], support vector machine regression (SVMR) [23], and radial basis function (RBF) neural network [24]. These retrieval models provide feasible solutions to disclose the linear and nonlinear relationship between reflectance and soil component contents. However, these methods rely on identifying indicative signatures from the spectral data curve's absorption positions, i.e., the extremum points (local minimum points), to retrieve the contents. Unfortunately, for spectral data, some special indicative signatures, especially for the weak absorption features, are not indicated as the extremum points. Actually, these weak absorptions always manifest as the oblique extremum points [25]. Thus, this study focuses on how to combine both the local minimum points and the oblique extremum points to extract indicative signatures and retrieve the SOM in black soil based on these combinatorial points from spectral data. Then, this study discusses and compares the retrieval performance of the radial basis function neural network with different indicative signature extraction methods. The experimental results show that this new idea of extracting indicative signatures for further retrieval is superior to traditional methods.

## 2. Materials and Methods

### 2.1. Study Area

The study area is situated in the western part of Qixing Farm of Jiansanjiang River, Fujin of Jiamusi City, Heilongjiang Province, in the northeast of China. It is located within latitude $47°07'$N to $47°23'$N, and longitude $132°36'$E to $132°46'$E, with a total area about approximately 300 km$^2$, which is a central black soil region in China (Figure 1). The Qixing Farm is located within the Three Rivers Plain, an extensive flat terrain that offers an ideal setting for large-scale mechanized farming operations. The soil surface is primarily composed of black, dark brown, and dark brown humus, with a granular structure. Below it lies a clayey sedimentary layer containing brown iron-manganese nodules and a brown-yellow clayey parent material underneath it. Black soil is known for its high fertility and is predominantly used in the cultivation of soybeans, rice, and corn.

### 2.2. Data Acquisition and Pre-Process

A total of 68 black soil samples, as shown in Figure 1, were collected from the study area. To eliminate errors caused by tillage and fertilizer application, S-shaped sampling was employed, avoiding certain special locations such as ridges and fertilizer accumulation areas. These surface soil samples, each weighing approximately 1 kg and collected at a depth of 10–20 cm, were first rid of impurities such as grains, weeds, and stones before being placed in cloth sample bags, numbered, and stored for processing. To remove any effects of moisture, particle size, or impurities in the soil sample on the spectral data, the samples were naturally air-dried and passed through a 120-mesh nylon screen. Following that, every 500 g of soil was subjected to hyperspectral detection and measurements of SOM content.

The spectrums of the samples were measured using an Analytical Spectral Devices (ASD) FieldSpec4 portable object spectrometer, while excluding the range of 350–400 nm and 2451–2500 nm due to interference from moisture and systematic errors associated with the instrument. The resulting data, consisting of 2050 spectral bands, were processed using the Savitzky-Golay (SG) smoothing method [26] to eliminate noise. Additionally, five nonlinear transformations were applied, namely reciprocal ($1/R$), reciprocal logarithmic ($\ln(1/R)$), square ($R^2$), cubic ($R^3$), and root mean square ($\sqrt{R}$), to the original spectrum to explore the indicative signatures further.

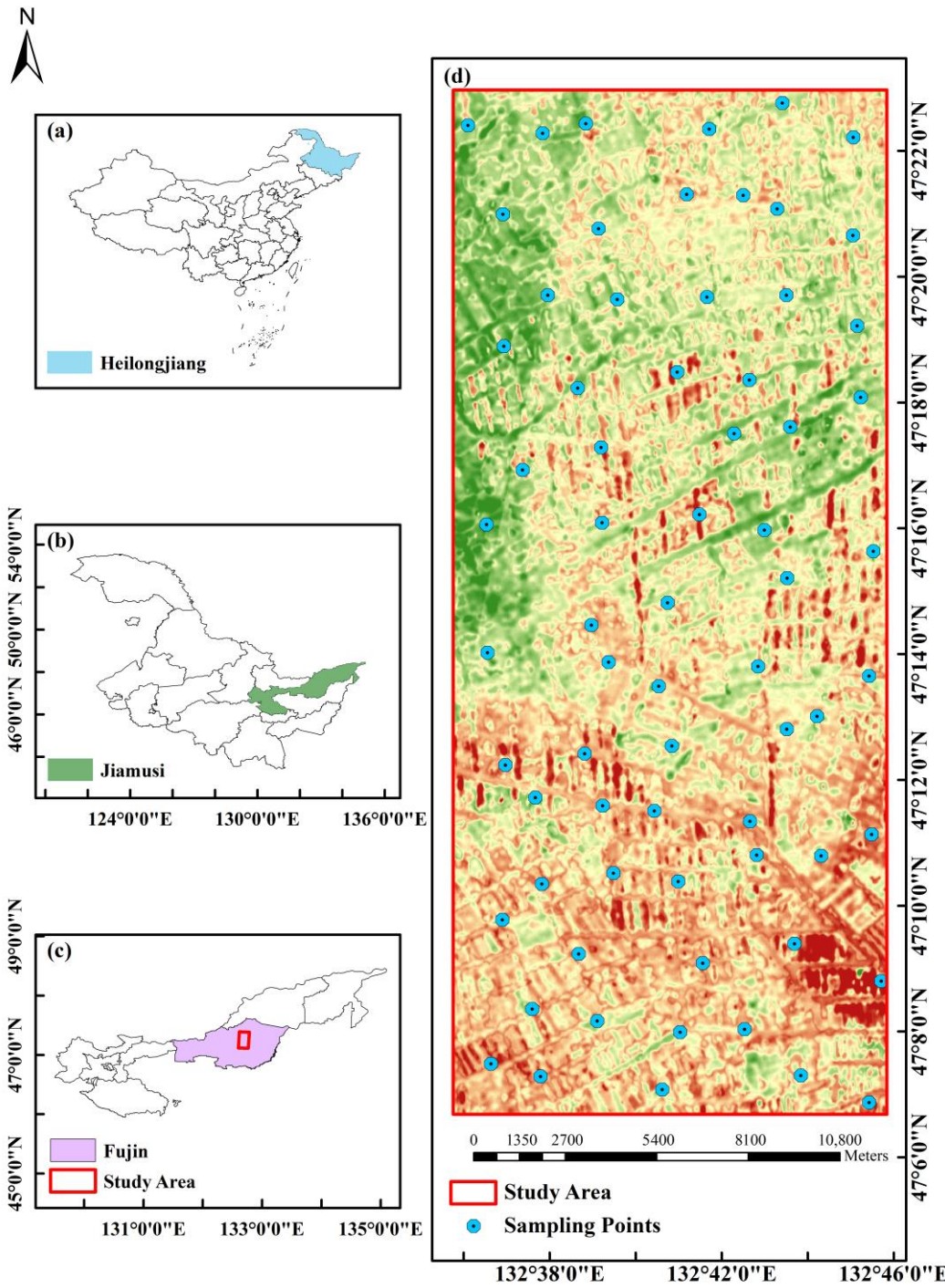

**Figure 1.** Geographic distribution of collected samples in the study area. (**a**) A map of China; (**b**) A map of Heilongjiang Province; (**c**) A map of Jiamusi City and study area; (**d**) The sample distribution in the study area.

The SOM contents of all samples were determined through wet oxidation at 180 °C, using a mixture of potassium dichromate and sulfuric acid. The 68 samples were sorted in ascending order according to their SOM contents and divided into 17 categories, with four samples in each category, which was designed in a 3-to-1 ratio for modeling and validation datasets. Three random samples were then selected from each category and assigned to the modeling dataset, leaving the remaining 17 samples for validation. Table 1 shows some statistical parameters of the SOM contents for all 68 samples.

**Table 1.** Statistical characteristics of SOM in soil samples from the study area.

| Dataset | Number of Samples | Minimum (g/kg) | Maximum (g/kg) | Mean (g/kg) | SD [1] (g/kg) | CV [2] (%) |
|---|---|---|---|---|---|---|
| Total dataset | 68 | 7.651 | 133.678 | 40.055 | 19.226 | 2.083 |
| Modeling dataset | 51 | 7.651 | 133.678 | 40.431 | 20.579 | 1.965 |
| Validation dataset | 17 | 12.408 | 76.433 | 38.929 | 14.923 | 2.609 |

[1] SD: Standard Deviation; [2] CV: Coefficient of Variation.

### 2.3. Indicative Signature Extraction

#### 2.3.1. Pearson Correlation Analysis

The Pearson coefficients defined in Equation (1) were calculated to analyze the relationship between the different spectral transformations and the SOM contents. In Equation (1), $j$ denoted the band number; $r_j$ was the correlation coefficient between the spectral data of soil samples in the $j^{th}$ band and its SOM contents; $n$ was the total number of the samples; $i$ denoted the soil sample number; $x_{ij}$ was the spectral data of the $i^{th}$ sample at the $j^{th}$ band; $\overline{x}_j$ was the average spectrum for all samples at the $j^{th}$ band; $y_i$ was the SOM content of the $i^{th}$ sample; and $\overline{y}$ denoted the average SOM content for all samples.

$$r_j = \frac{\sum\limits_{i=1}^{n}(x_{ij} - \overline{x}_j)(y_i - \overline{y})}{\sqrt{\sum\limits_{i=1}^{n}(x_{ij} - \overline{x}_j)^2 \sum\limits_{i=1}^{n}(y_i - \overline{y})^2}}. \tag{1}$$

#### 2.3.2. Extremum Method

Commonly, the indicative signatures of the spectrum are selected with correlation analysis [6,14,27]. Although the correlation analysis can correlate well between reflectance and sample contents, the selected results lack physical significance because the indicative signatures are always presented as the absorption bands or points, not the maximum correlation coefficient bands. Thus, a combination of the absorption bands and the correlation coefficient analysis is usually used to extract the indicative signatures from the spectrum. In this study, this combination is called as the extremum method. Unfortunately, not all indicative signatures appeared as the extremum points, especially for some weak absorption features. So, how to extract this type of points is an interesting issue for the following retrieval process. In this study, we have improved a method to extract these signatures and subsequently retrieve the contents of SOM in black soil by utilizing the new extracted signatures. First, we will introduce the improved method, followed by a discussion and comparison of the results.

#### 2.3.3. Oblique Extremum Method and Improved Oblique Extremum Method

Here, the oblique extremum points [25] are used to define this kind of special absorption phenomenon, in which absorptions are not indicated as the extremum points of the spectrum. To illustrate the detail, consider the signal defined in Equation (2), below.

$$y(t) = t + 0.9\sin(t), \ t \in [1, 64]. \tag{2}$$

As shown in Figure 2a, obviously, the signal $y(t)$ is strictly monotonically increasing, without any extremum points. However, intuitively, there indeed exist some special signatures in this signal. To address this problem, Yang et al. [25] defined the oblique extremum points to characterize these special points and proposed an approach to detect them.

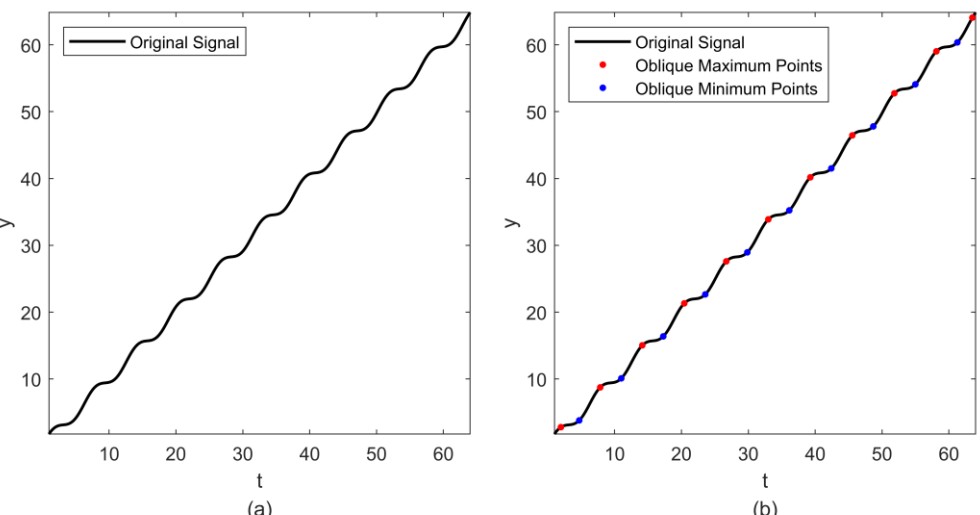

**Figure 2.** The original signal $y(t)$ (**a**) and its oblique extremum points (**b**).

Given a signal $y(t)$, let $(t_a, f(t_a))$ and $(t_b, f(t_b))$ $(t_a < t_b)$ be two consecutive inflection points of $y(t)$, and $l(t)$ to be the straight line connecting these two inflection points. Denote

$$y_l(t) = y(t) - l(t), t \in [t_a, t_b],\qquad(3)$$

then, the local maximum (or minimum) point $\xi$ of $y_l(t)$ is called an oblique local maximum (or minimum) point of $y(t)$.

Therefore, the key to extracting the oblique extremum points lies in identifying the inflection points of the signal. As described in [25], the first-order derivative of the original signal is used to detect the inflection points. The adjacent inflection points generate a line segment that forms the trend curve of the original signal. Thus, the oblique extremum points are determined as the extremum points of the subtraction result of the trend curve from the original signal. Figure 2b demonstrates that the extracted oblique extremum points of $y(t)$ are all located as the gentle humps in the curve. These results in Figure 2b indicate that certain special signatures in remote sensing imagery and spectral data can also be identified as the oblique extremum points. Before we introduce the oblique extremum points in spectral data, let us first discuss the algorithm for detecting such points.

As described in the literature [25], the determination of the inflection points of a signal is crucial in defining the oblique extremum point and implementing the algorithm to extract them. Yang et al. [25] employed the first-order derivative to do that. However, in theory, the inflection point of a continuous signal is defined as the demarcation point between the concave and the convex intervals of the signal. The second-order derivative is more reasonable than the first-order derivative for detecting the inflection points of a signal. Therefore, we develop an improved algorithm for extracting oblique extremum points from a signal $y(x(t))$ based on the second-order derivative. Details on this algorithm can be found in Algorithm 1.

---

**Algorithm 1** Improved algorithm to extract oblique extremum points.

---

1. Calculate the second-order derivative of original hyperspectral signal $y(x(t))$

$$
\begin{aligned}
y''(x(t)) &= \frac{y'(x(t+1/2)) - y'(x(t-1/2))}{x(t+1/2) - x(t-1/2)} = \frac{\frac{y(x(t+1)) - y(x(t))}{x(t+1) - x(t)} - \frac{y(x(t)) - y(x(t-1))}{x(t) - x(t-1)}}{x(t+1/2) - x(t-1/2)}, \\
&= \frac{y(x(t+1)) + y(x(t-1)) - 2y(x(t))}{(\Delta x)^2}, \ t = 2, \ldots, end - 1
\end{aligned}
\qquad(4)
$$

---

---

**Algorithm 1** *Cont.*

---

2. Extract the oblique maximum points of the signal.
Step 1. Define and identify the $k^{th}$ concave interval

$$
I_{kh}^{CC} = [x(i), \ldots, x(i+h-1)], \ k = 1, \ldots, m
$$
$$
s.t. \begin{cases} y''(x(t)) < 0, \ t = i, \ldots, i+h-1, \\ \qquad y''(x(i-1)) \geq 0, \\ \qquad y''(x(i+h)) \geq 0, \\ \qquad 2 \leq i < end-1, \\ \qquad h \geq 1, \end{cases} \tag{5}
$$

in which $h$ is the interval length,
$m$ is the total number of concave intervals, and the superscript $CC$ represents concave.
Step 2. Find inflection points on both sides of the concave interval $I_{kh}^{CC}$, and
construct the trend line $l_{kh}^{CC}(x(t))$.

a.    Since the first point $x(i)$ of $I_{kh}^{CC}$ is a concave point, so the previous point $x(i-1)$ must be a
convex point or an inflection point, according to Equation (5). If $x(i-1)$ is a convex point,
not an inflection point, then the abscissa and ordinate of the inflec-tion point at the left end
of $I_{kh}^{CC}$ is defined as the mean value of these two adjacent points in Equation (6)

$$
x_{CC}^{left\_inf} = \frac{x(i-1)+x(i)}{2}, y_{CC}^{left\_inf} = \frac{y(i-1)+y(i)}{2}, \tag{6}
$$

else  if $x(i-1)$ is an inflection point, then set

$$
x_{CC}^{left\_inf} = x(i-1), y_{CC}^{left\_inf} = y(i-1), \tag{7}
$$

where the superscript $left\_inf$ represents the left inflection point.
b.    Similarly, the last point $x(i+h-1)$ of $I_{kh}^{CC}$ is a concave point, so the next point
$x(i+h)$ must be a convex point or an inflection point. If $x(i+h)$ is a convex point, define
abscissa and ordinate of the inflection point at the right end of $I_{kh}^{CC}$ as

$$
x_{CC}^{right\_inf} = \frac{x(i+h-1)+x(i+h)}{2}, y_{CC}^{right\_inf} = \frac{y(i+h-1)+y(i+h)}{2}, \tag{8}
$$

else

$$
x_{CC}^{right\_inf} = x(i+h), y_{CC}^{right\_inf} = y(i+h), \tag{9}
$$

where the superscript $right\_inf$ represents the right inflection point.
c.    Connect the left and right inflection points of the concave interval $I_{kh}^{CC}$ ($k = 1, \ldots, m$) with
a straight line $l_{kh}^{CC}(x(t))$ as the trend curve of $I_{kh}^{CC}$.

Step 3. Subtract the trend line from the original signal: $y_{l_{kh}^{CC}}(x(t)) = y(x(t)) - l_{kh}^{CC}(x(t))$, where
$x(t) \in I_{kh}^{CC}$.
Step 4. Obtain all oblique maximum points $x_{kh}^{OMAX}$ and construct a set

$$
U^{OMAX} = \bigcup_{k=1,2,\ldots,m} x_{kh}^{OMAX} = \bigcup_{k=1,2,\ldots,m} \text{argmax}\left( \left| y_{l_{kh}^{CC}}(x(t)) \right| \right), \tag{10}
$$

in which the superscript $OMAX$ represents oblique maximum.

---

---

**Algorithm 1** *Cont.*

---

3. Extract the oblique minimum points of the signal.
Step 1. Define and identify the $k^{th}$ convex interval

$$I_{kl}^{CV} = [x(i), \ldots, x(i+l-1)], \ k = 1, \ldots, n$$
$$s.t. \begin{cases} y''(x(t)) > 0, \ t = i, \ldots, i+l-1, \\ \qquad y''(x(i-1)) \leq 0, \\ \qquad y''(x(i+l)) \leq 0, \\ \qquad 2 \leq i < end - 2, \\ \qquad l \geq 1, \end{cases} \qquad (11)$$

in which $l$ is the interval length, $n$ is the total number of convex intervals, and the superscript $CV$ represents convex.
Step 2. Find inflection points on both sides of the convex interval $I_{kl}^{CV}$, and constructthe trend line $l_{kl}^{CV}(x(t))$.

a.   Since the first point $x(i)$ of $I_{kl}^{CV}$ is a convex point, the previous point $x(i-1)$ must be a concave point or an inflection point, according to Equation (11). If $x(i-1)$ is a concave point, define abscissa and ordinate of the inflection point at the left end of $I_{kl}^{CV}$ as

$$x_{CV}^{left\_inf} = \frac{x(i-1) + x(i)}{2}, y_{CV}^{left\_inf} = \frac{y(i-1) + y(i)}{2}, \qquad (12)$$

else

$$x_{CV}^{left\_inf} = x(i-1), y_{CV}^{left\_inf} = y(i-1), \qquad (13)$$

where the superscript $left\_inf$ represents the left inflection point.
b.   Similarly, the last point $x(i+l-1)$ of $I_{kl}^{CV}$ is a convex point, so the next point $x(i+l)$ must be a concave point or an inflection point. If $x(i+l)$ is a concave point, define abscissa and ordinate of the inflection point at the right end of $I_{kl}^{CV}$ as

$$x_{CV}^{right\_inf} = \frac{x(i+l-1) + x(i+l)}{2}, y_{CV}^{right\_inf} = \frac{y(i+l-1) + y(i+l)}{2}, \qquad (14)$$

else

$$x_{CV}^{right\_inf} = x(i+l), y_{CV}^{right\_inf} = y(i+l), \qquad (15)$$

where the superscript $right\_inf$ represents the right inflection point.
c.   Connect the left and right inflection points of convex interval $I_{kl}^{CV}$ $(k = 1, \ldots, n)$ with a straight line $l_{kl}^{CV}(x(t))$ as the trend line of $I_{kl}^{CV}$.
Step 3. Subtract the trend line from the original signal : $y_{l_{kl}^{CV}}(x(t)) = y(x(t)) - l_{kl}^{CV}(x(t))$, where $x(t) \in I_{kl}^{CV}$.
Step 4. Obtain all oblique minimum points $x_{kl}^{OMIN}$ and construct a set

$$U^{OMIN} = \bigcup_{k=1,2,\ldots,n} x_{kl}^{OMIN} = \bigcup_{k=1,2,\ldots,n} argmax\left(\left|y_{l_{kl}^{CV}}(x(t))\right|\right), \qquad (16)$$

in which the superscript $OMIN$ represents oblique minimum.

---

Let's consider the spectral data of a measured soil sample after SG smoothing as an example, to further illustrate the difference between the extremum method and the improved oblique extremum method for extracting the indicative signatures. As shown in Figure 3a, for the extremum method, only 30 local minimum points—also known as the absorption points—were extracted, with no local minimum points detected in 600–1700 nm. In other words, almost all the absorption signatures are located on either side of the signal. However, with the improved oblique extremum method, 133 points that covered the entire wavelength of the signal are extracted totally. Clearly, the extracted results of the improved oblique extremum method reflect more information and detail about the signal than the extremum method does. Moreover, the improved oblique extremum method has the ability

to extract potential weak absorption points in the spectral data. The comparative details are all demonstrated in Figure 3.

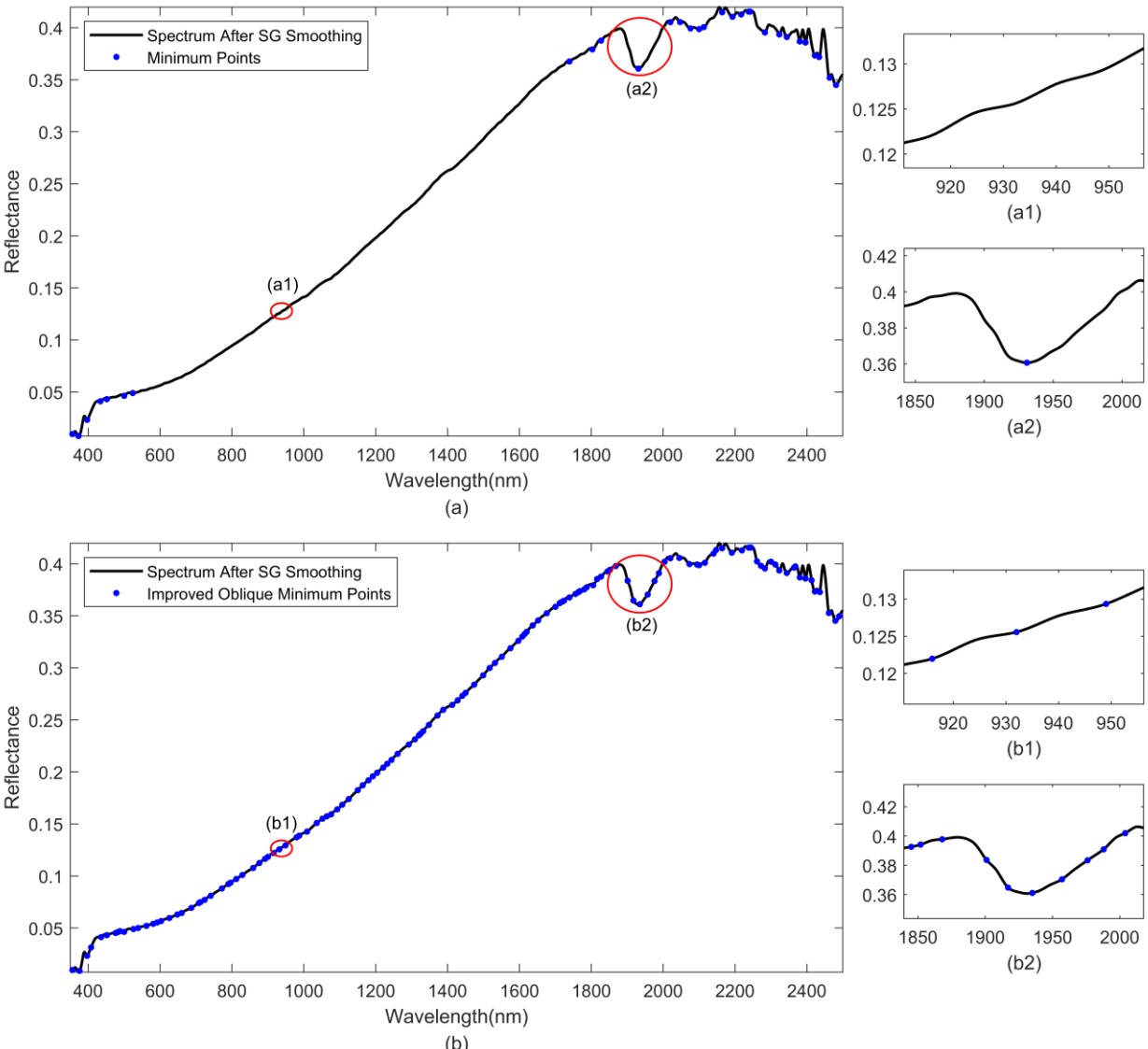

**Figure 3.** Extracted results of minimum points with the extremum method and oblique minimum points with improved oblique extremum method. (**a**) Minimum points extracted with extremum method; (**a1**) and (**a2**) are partial enlargement of (**a**); (**b**) Oblique minimum points extracted with improved oblique extremum method; (**b1**) and (**b2**) are partial enlargement of (**b**).

### 2.4. Retrieval Method

In order to explore the nonlinear relationship between spectral transformations and SOM contents, a nonlinear regression model, radial basis function neural network, is utilized in this study. The RBF neural network is a feed-forward neural network with three layers: input layer, hidden layer, and output layer [24]. The input layer receives the training data. The hidden layer applies an activation function to perform a nonlinear transformation on the input data, where the Gaussian radial basis function is frequently utilized. The activation function for the output layer is the linear function. The final output is obtained by taking the linear weighted sum of the outputs of the neurons in the hidden layer. Therefore, the RBF neural network boasts a simpler structure, faster learning convergence speed, and better generalization ability in comparison to other machine learning algorithms [27].

## 3. Results and Discussion

### 3.1. The Spectral Characteristics Analysis

The original and denoised spectra of all the samples are demonstrated in Figure 4. Generally, the minor differences of the reflectance play a crucial role in revealing the undiscovered relationships between the SOM contents and the reflectance of the samples.

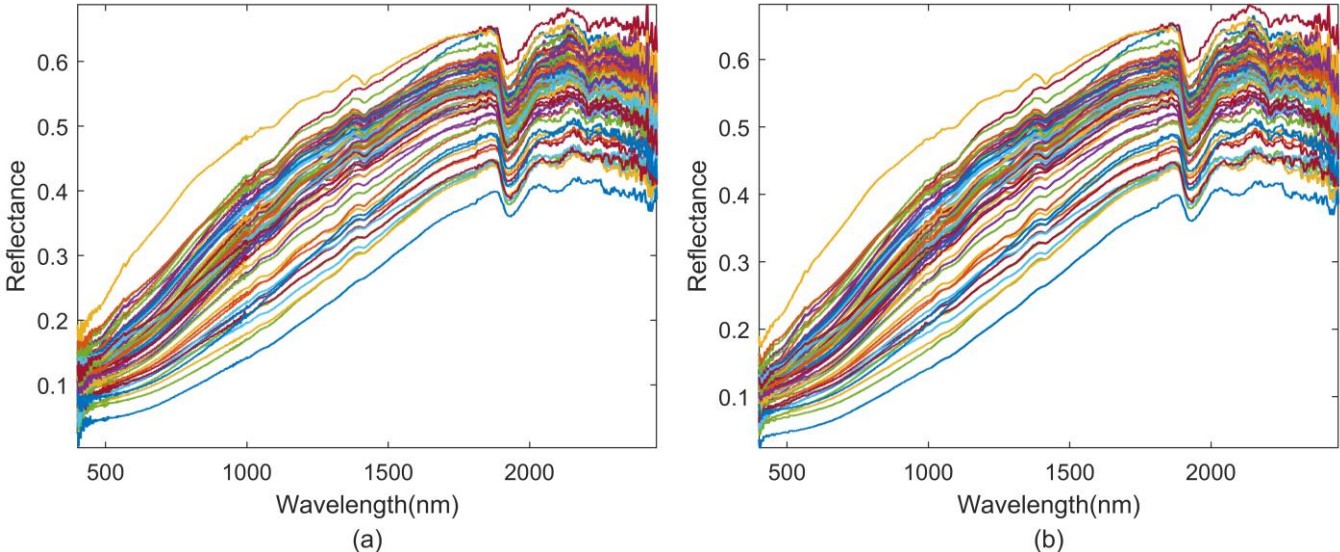

**Figure 4.** Spectral reflectance of 68 samples. (**a**) The original spectra; (**b**) SG smoothing results.

To gain a deeper understanding of the spectral characteristics and investigate the relationship between soil reflection spectra and SOM content, all the 68 samples were clustered into four classes using the K-means clustering method [28] (Figure 5). The number of the classes was determined with the Elbow method [29]. Based on Figure 5, it is apparent that there exists a negative correlation between soil spectral reflectance and SOM content, meaning that the higher the SOM content, the lower the spectral reflectance. Obviously, the significant absorption features are located near 1400 nm and 1900 nm, primarily due to the stretching vibration of hydroxyl and the bending vibration of Al-OH of the water molecules [12,30,31]. These features are also caused by the water vapor in the air during the laboratory spectroscopy measurements. Additionally, an absorption close to 2200 nm, a characteristic of kaolinite, can also be observed, resulting from the combination of Al-OH bend and O-H stretch [32].

### 3.2. Extraction Results of Indicative Signatures for SOM

3.2.1. Correlation Analysis between the SOM Contents and the Spectral Reflectance and Their Different Transformations

The correlation coefficients between the SOM contents of the samples and their spectral reflectance, as well as their five spectral transformations ($1/R$, $\ln(1/R)$, $R^2$, $R^3$, and $\sqrt{R}$), were calculated and illustrated in Figure 6. There is a strong negative correlation between the SOM contents and the original spectral reflectance within the range of 401–2450 nm, which is consistent with the findings of the clustering analysis discussed in Section 3.1. Regarding the five spectral transformations, $R^2$, $R^3$, and $\sqrt{R}$ coincide with the original spectral reflectance $R$, whereas the transformations $1/R$ and $\ln(1/R)$ exhibit a mirror symmetry, as depicted in Figure 6a. From Figure 6, it can be observed that: (1) the coefficients between the original spectral reflectance and SOM contents fall within the range of $[-0.615, -0.336]$; and (2) the transformations $1/R$, $\ln(1/R)$, and $\sqrt{R}$ can marginally improve the correlation between the original spectral reflectance and the SOM content.

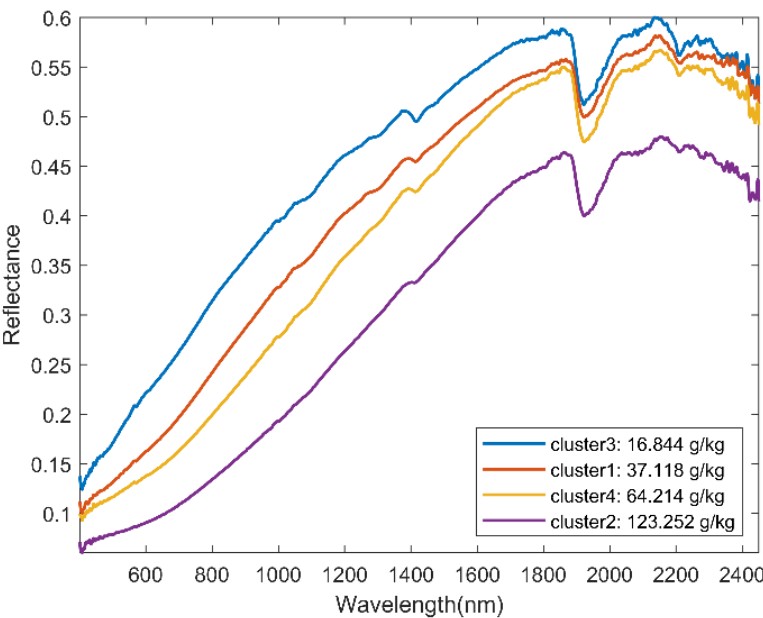

**Figure 5.** Mean values of the spectral reflectance for four different categories after clustering.

Then, the indicative bands were chosen based on the correlation analysis results, guided by two principles: the first is that the selected bands must have higher correlation coefficients than others, and the second is that they should be separated from each other; in other words, they should not be adjacent. All the selected indicative bands are illustrated in Table 2, including the peaks of the correlation coefficients and corresponding bands. It is worth noting that most of the indicative bands extracted by the correlation analysis are situated within the range of 600–1000 nm, which is basically consistent with the finding of Yuan et al. [33].

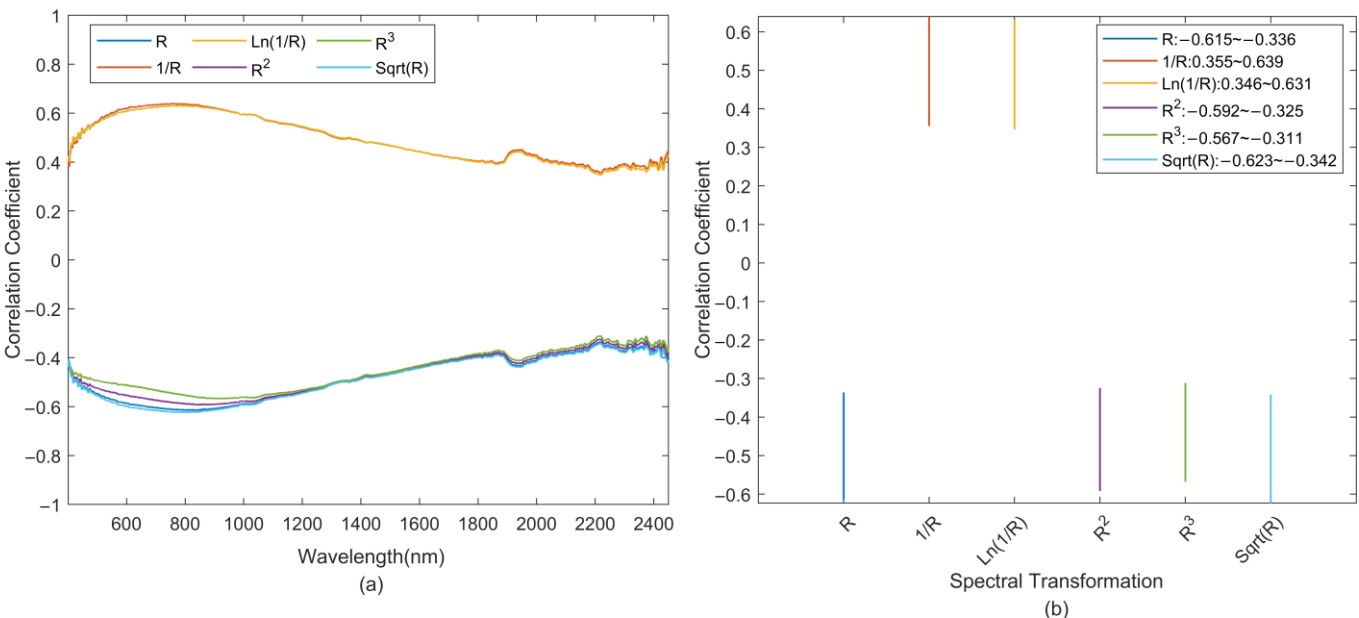

**Figure 6.** Correlation coefficients of the SOM contents and the original reflectance, as well as their five spectral transformations. (**a**) Correlation coefficients; (**b**) Range of correlation coefficients.

**Table 2.** Indicative bands and correlation coefficients between SOM contents and different spectral transformations.

| Spectral Transformation | Indicative Bands (nm) | Correlation Coefficients |
|:---:|:---:|:---:|
| $R$ | 838 | −0.607 |
| | 1142 | −0.573 |
| | 537 | −0.546 |
| | 1442 | −0.521 |
| $1/R$ | 762 | 0.643 |
| | 838 | 0.642 |
| | 697 | 0.638 |
| $\ln(1/R)$ | 838 | 0.629 |
| | 683 | 0.616 |
| | 1003 | 0.608 |
| $R^2$ | 929 | −0.584 |
| | 1280 | −0.541 |
| | 574 | −0.519 |
| $R^3$ | 1042 | −0.563 |
| | 941 | −0.558 |
| | 1145 | −0.552 |
| $\sqrt{R}$ | 838 | −0.619 |
| | 1141 | −0.576 |
| | 537 | −0.562 |

### 3.2.2. Indicative Signatures Extracted by Extremum Method

For remote sensing imagery or spectral data, the absorption points are represented as the local minimum points in the spectral curve, and the indicative signatures are always extracted from these absorption points. Thus, the extremum method to extract the indicative signatures here means that the characteristic bands are extracted from the local minimum points of the spectral curve based on their corresponding correlation results. For this case data, the extracted results are demonstrated in Table 3.

**Table 3.** Indicative signatures obtained by extremum method with different spectral transformations.

| Spectral Transformation | Indicative Bands (nm) | Correlation Coefficients |
|:---:|:---:|:---:|
| $R$ | 408 | −0.477 |
| | 1773 | −0.467 |
| | 998 | −0.595 |
| $1/R$ | 2354 | 0.474 |
| | 463 | 0.544 |
| | 1002 | 0.612 |
| $\ln(1/R)$ | 2354 | 0.463 |
| | 463 | 0.533 |
| | 1002 | 0.607 |
| $R^2$ | 408 | −0.464 |
| | 1773 | −0.461 |
| | 998 | −0.579 |
| $R^3$ | 408 | −0.441 |
| | 1773 | −0.453 |
| | 998 | −0.559 |
| $\sqrt{R}$ | 408 | −0.478 |
| | 1773 | −0.469 |
| | 998 | −0.601 |

Table 3 reveals an intriguing observation: that the extracted results for transformations $R$, $R^2$, $R^3$, and $\sqrt{R}$ are the same, i.e., 408 nm, 998 nm, and 1773 nm. For the transformations $1/R$ and $\ln(1/R)$, the results are the same, too, i.e., 463 nm, 1002 nm, and 2354 nm. The correlation results between the spectral data and SOM contents for these five transformations, namely $1/R$, $\ln(1/R)$, $R^2$, $R^3$, and $\sqrt{R}$, exhibit slight variations, suggesting that spectral transformations can marginally affect the correlation coefficients.

### 3.2.3. Indicative Signatures Extracted by Improved Oblique Extremum Method

As described in Section 2.3.3, the oblique minimum points in the spectral curve may hint at potential weak absorptions. Thus, we proposed a new idea to extract the indicative signatures based on oblique extremum points with the improved oblique extremum method. More specifically, the indicative signatures are derived from oblique minimum points, which encompasses all the local minimum points [25]. All the extracted indicative signatures are illustrated in Table 4. By utilizing oblique minimum points as the extracted indicative signatures, all absorption points of the spectral data are accounted for, and it also allows for the identification of potential weak absorptions depicted as oblique extremum points rather than local minimum points. Further, in order to compare the retrieval results based on oblique minimum points, another indicative signature that only contains the oblique minimum points without the local minimum points is also considered. Here, oblique minimum without local minimum means that the local minimum points of the spectral curve are deleted from the extracted oblique minimum points.

**Table 4.** Indicative signatures obtained by improved oblique extremum method with different spectral transformations.

| Spectral Transformation | Indicative Signatures Kind | Bands (nm) | Correlation Coefficients |
|---|---|---|---|
| $R$ | Oblique Minimum | 841 | −0.607 |
| | | 951 | −0.601 |
| | | 605 | −0.567 |
| | Oblique Minimum Without Local Minimum | 841 | −0.607 |
| | | 951 | −0.601 |
| | | 701 | −0.588 |
| | | 605 | −0.567 |
| $1/R$ | Oblique Minimum | 863 | 0.636 |
| | | 703 | 0.637 |
| | | 941 | 0.624 |
| | | 557 | 0.605 |
| | Oblique Minimum Without Local Minimum | 863 | 0.636 |
| | | 1587 | 0.494 |
| | | 557 | 0.605 |
| $\ln(1/R)$ | Oblique Minimum | 863 | 0.624 |
| | | 703 | 0.617 |
| | | 941 | 0.617 |
| | | 557 | 0.583 |
| | Oblique Minimum Without Local Minimum | 863 | 0.624 |
| | | 1587 | 0.494 |
| $R^2$ | Oblique Minimum | 841 | −0.577 |
| | | 415 | −0.486 |
| | | 1516 | −0.503 |
| | Oblique Minimum Without Local Minimum | 841 | −0.577 |
| | | 605 | −0.525 |
| | | 1516 | −0.503 |
| | | 1677 | −0.474 |

**Table 4.** *Cont.*

| Spectral Transformation | Indicative Signatures Kind | Bands (nm) | Correlation Coefficients |
|---|---|---|---|
| $R^3$ | Oblique Minimum | 841 | −0.542 |
| | | 419 | −0.470 |
| | Oblique Minimum Without Local Minimum | 841 | −0.542 |
| | | 951 | −0.558 |
| | | 605 | −0.476 |
| | | 701 | −0.503 |
| $\sqrt{R}$ | Oblique Minimum | 841 | −0.618 |
| | | 951 | −0.609 |
| | | 605 | −0.584 |
| | Oblique Minimum Without Local Minimum | 841 | −0.618 |
| | | 951 | −0.609 |
| | | 701 | −0.604 |

*3.3. The Retrieval Results Analysis with Different Indicative Signature Extraction Methods*

The determination coefficient ($R^2$), root mean of squared error (RMSE), and the ratio of the performance to deviation (RPD) were used to evaluate the performance of the retrieval results. These three evaluation criteria were defined in Equations (17)–(19), respectively. Specifically, $R^2$ acted as the criterion to assess the effectiveness of the retrieval model. Commonly, the closer the value is to 1, the better the fitting results. RMSE is used to measure the accuracy of the retrieval model, and a smaller value of RMSE indicates higher accuracy. Further, the value of RPD always represents the predictive ability of the model, and a higher value signifies a better prediction result. Therefore, in general, high $R^2$ and high RPD, but low RMSE, are characteristics of an excellent retrieval model. As pointed out by Chang et al. [34], for $R^2 > 0.5$, if RPD $< 1.4$, the model fails to predict the samples; if $1.4 \leq$ RPD $< 2.0$, the model can provide a rough estimation of the samples; and if RPD $\geq 2.0$, the model exhibits excellent predictive ability.

$$R^2 = \frac{\sum\limits_{i=1}^{n} (\hat{y}_i - \overline{y})^2}{\sum\limits_{i=1}^{n} (y_i - \overline{y})^2}, \tag{17}$$

$$RMSE = \sqrt{\frac{\sum\limits_{i=1}^{n} (y_i - \hat{y}_i)^2}{n}}, \tag{18}$$

$$RPD = \frac{SD_v}{RMSE_v}. \tag{19}$$

In Equations (17)–(19), $i$ was the soil sample number; $n$ denoted the total number of samples; $y_i$ and $\hat{y}_i$ were the observed and predicted value of the $i^{th}$ sample, respectively; $\overline{y}$ was the mean of the observed data; and $SD_v$ and $RMSE_v$ were the standard deviation and the root mean of the squared error of the validation dataset, respectively.

3.3.1. The Retrieval Results Based on Indicative Signatures with Correlation Analysis

The evaluation results of the RBF neural network based on indicative signatures using correlation analysis are all illustrated in Table 5. The results indicate that: (1) the retrieval model relying on correlation analysis can only provide a rough estimation, given that the values of $R_m^2$ and $R_v^2$ exceed 0.5, and RPD ranges between 1.4 to 2.0 for both modeling and validation datasets; (2) among the six spectral transformations, original reflectance $R$ displays the best-fitting performance for the modeling dataset, with the highest being

$R_m^2 = 0.659$ and the smallest being $\text{RMSE}_m = 11.900$ g/kg; (3) $\ln(1/R)$ yields the most favorable prediction outcome, with an RPD of 1.885. Overall, the transformation $\ln(1/R)$ appears to be more appropriate for retrieving the SOM contents when utilizing correlation analysis to identify indicative signatures.

**Table 5.** RBF neural network model performance results to retrieve the SOM contents based on indicative signatures extracted by correlation analysis.

| Spectral Transformation | Modeling Dataset | | | Validation Dataset | | |
|:---:|:---:|:---:|:---:|:---:|:---:|:---:|
| | $R_m^2$ | $\text{RMSE}_m$ (g/kg) | $R_v^2$ | $\text{RMSE}_v$ (g/kg) | RPD | |
| $R$ | 0.659 | 11.900 | 0.730 | 8.755 | 1.705 | |
| $1/R$ | 0.534 | 13.908 | 0.639 | 8.025 | 1.860 | |
| $\ln(1/R)$ | 0.653 | 12.004 | 0.602 | 7.919 | 1.885 | |
| $R^2$ | 0.579 | 13.219 | 0.725 | 8.126 | 1.836 | |
| $R^3$ | 0.527 | 14.025 | 0.751 | 8.452 | 1.766 | |
| $\sqrt{R}$ | 0.638 | 12.258 | 0.562 | 8.773 | 1.701 | |

### 3.3.2. The Retrieval Results Based on Indicative Signatures with Extremum Method

The retrieval results based on the extremum method demonstrated in Table 6 indicate that: (1) only one model with $R^2$ transformation can be used to retrieve the contents of the SOM with $R_m^2 = 0.550 > 0.5$, $R_v^2 = 0.702 > 0.5$, and RPD $= 1.643 > 1.4$; (2) although the transformation $R^3$ retrieval model based on minimum indicative signatures achieved a good predictive effect with RPD $= 1.818 > 1.643$, it did not fit well with the modeling dataset with $R_m^2 = 0.465 < 0.5$; (3) combining the results listed in Table 3, the indicative signatures of four transformations, $R$, $R^2$, $R^3$ and $\sqrt{R}$, are the same, but the retrieval performances for them, as showcased in Table 6, varied significantly, indicating that the transformations $R^2$ are helpful to improve the SOM predictive result.

**Table 6.** RBF neural network model performance results to retrieve the SOM contents based on indicative signatures extracted by extremum method.

| Spectral Transformation | Modeling Dataset | | | Validation Dataset | | |
|:---:|:---:|:---:|:---:|:---:|:---:|:---:|
| | $R_m^2$ | $\text{RMSE}_m$ (g/kg) | $R_v^2$ | $\text{RMSE}_v$ (g/kg) | RPD | |
| $R$ | 0.583 | 13.163 | 0.586 | 10.679 | 1.397 | |
| $1/R$ | 0.498 | 14.441 | 0.432 | 9.785 | 1.525 | |
| $\ln(1/R)$ | 0.518 | 14.145 | 0.337 | 11.726 | 1.273 | |
| $R^2$ | 0.550 | 13.663 | 0.702 | 9.085 | 1.643 | |
| $R^3$ | 0.465 | 14.898 | 0.742 | 8.208 | 1.818 | |
| $\sqrt{R}$ | 0.599 | 12.896 | 0.541 | 11.934 | 1.251 | |

### 3.3.3. The Retrieval Results Based on Indicative Signatures with Improved Oblique Extremum Method

The retrieval results based on indicative signatures extracted by the improved oblique extremum method are all illustrated in Table 7. It is exciting that the retrieval model based on oblique minimum without the minimum performs better than oblique minimum for all six transformations, as compared in Table 7. Further, from the view of the RPD performance criterion, all six models are suitable for SOM retrieval with $R_m^2 > 0.5$, $R_v^2 > 0.5$, and RPD $> 1.4$. Four of the models yield outstanding predictions, with RPD $> 2.0$. This result reveals that the oblique minimum (non-minimum) signatures of soil spectra are crucial for retrieving SOM content, which is a new insight never reported in previous studies.

**Table 7.** RBF neural network model performance results to retrieve the SOM contents based on indicative signatures extracted by improved oblique extremum method.

| Spectral Transformation | Indicative Signatures Kind | Modeling Dataset | | Validation Dataset | | |
|---|---|---|---|---|---|---|
| | | $R_m^2$ | $RMSE_m$ (g/kg) | $R_v^2$ | $RMSE_v$ (g/kg) | RPD |
| $R$ | Oblique Minimum | 0.642 | 12.185 | 0.660 | 8.692 | 1.717 |
| | Oblique Minimum Without Local Minimum | 0.706 | 11.046 | 0.747 | 6.280 | 2.376 |
| $1/R$ | Oblique Minimum | 0.624 | 12.502 | 0.736 | 7.715 | 1.934 |
| | Oblique Minimum Without Local Minimum | 0.598 | 12.916 | 0.745 | 6.751 | 2.211 |
| $\ln(1/R)$ | Oblique Minimum | 0.622 | 12.532 | 0.501 | 9.440 | 1.581 |
| | Oblique Minimum Without Local Minimum | 0.550 | 13.662 | 0.591 | 7.677 | 1.944 |
| $R^2$ | Oblique Minimum | 0.560 | 13.512 | 0.738 | 10.625 | 1.405 |
| | Oblique Minimum Without Local Minimum | 0.614 | 12.655 | 0.842 | 6.652 | 2.243 |
| $R^3$ | Oblique Minimum | 0.416 | 15.577 | 0.515 | 10.096 | 1.478 |
| | Oblique Minimum Without Local Minimum | 0.547 | 13.721 | 0.744 | 8.639 | 1.727 |
| $\sqrt{R}$ | Oblique Minimum | 0.656 | 11.946 | 0.668 | 8.382 | 1.780 |
| | Oblique Minimum Without Local Minimum | 0.671 | 11.685 | 0.772 | 6.642 | 2.247 |

Upon comparing the results displayed in Tables 5 and 6, it is clear that the RBF neural network based on improved oblique extremum method outperforms both correlation analysis and extremum methods. Specifically, for original reflectance $R$, the RPD value of oblique minimum without the minimum retrieval model is 2.376, whereas the RPD values for the correlation analysis and extremum methods are 1.705 and 1.397, respectively.

It is interesting to compare the correlation coefficients of the indicative bands obtained by the improved oblique extremum method with those extracted with only the correlation analysis. As illustrated in Tables 2 and 4, the correlation coefficients of the indicative bands obtained by the improved oblique extremum method are not the highest. Thus, relying on only the correlation analysis to extract the indicative signatures seems to be unreasonable because the correlation coefficient can only disclose the linear relationship between reflectance and SOM contents. Actually, the relationship is always nonlinear, as the results show in Figure 6.

### 3.3.4. Comparison of Retrieval Results with Different Indicative Signature Extraction Methods

The performance comparison of the optimal models for six transformations with different indicative signature extraction methods are demonstrated in Figure 7. The results show that, on the whole, models based on indicative signatures extracted through the improved oblique extremum method outperform both correlation analysis and extremum methods; they have higher $R^2$, lower RMSE, and higher RPD. These results reveal the fact that the improved oblique extremum method is capable of detecting the possible weak indicative signatures in the spectral data, which were hidden as the oblique minimum points rather than local minimum points. Further, these weak indicative signatures are of great importance in subsequent retrieval processes.

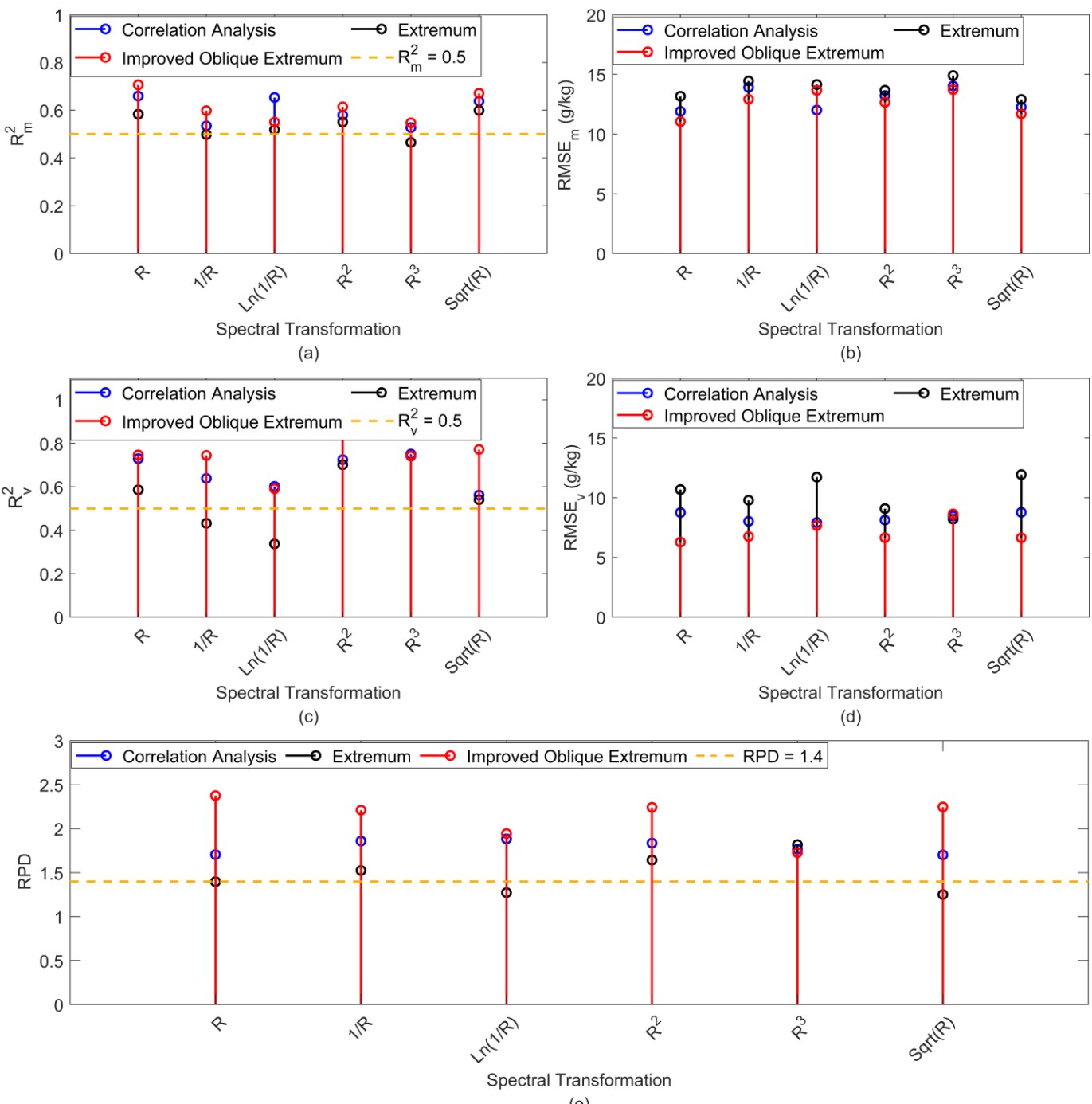

**Figure 7.** Comparison of the fitting results ($R_m^2$ and $R_v^2$), modeling accuracy ($RMSE_m$ and $RMSE_v$), and predictive property (RPD) between the optimal models for six transformations with different indicative signature extraction methods. (**a**,**b**) are the comparison results of $R_m^2$ and $RMSE_m$ for modeling dataset, respectively; (**c**–**e**) are the comparison results of $R_v^2$, $RMSE_v$, RPD for validation dataset, respectively.

## 4. Conclusions

This study focuses on how to extract the indicative signatures from the spectral data during the retrieval process based on remote sensing technique. Herein, we provide a new insight into the indicative signatures as the oblique extremum points of the data. The oblique extremum points include not just the traditional local minimum points, also known as absorption points, but also potential weak absorption points. However, these weak



absorption points are not presented as local minimum points of the signal. The results from a case study of the retrieval of SOM contents in black soil located in Northeast China indicate the effectiveness of this new insight. Thus, the oblique extremum points of the spectral data are suggested to be the extracted indicative signatures for further retrieval models. Commonly, the relationship between reflectance or their transformations and SOM contents is nonlinear, as indicated by the correlation analysis. In other words, the correlation coefficients between the SOM contents and the reflectance or their transformations reveal a nonlinear relationship in nature. Therefore, a retrieval model with nonlinear mapping capability, such as an RBF neural network model, is more appropriate than those that can only disclose linear relationships. Another important fact to note is that the spectral transformations used in this paper, namely the reciprocal, reciprocal logarithmic, square, cubic, and root mean square transformations, do not alter the indicative signatures extracted by the extremum method. However, it is worth mentioning that the square transformation improved the retrieval performance of the SOM. Therefore, identical indicative signatures are obtained in different spectral transformations, and it is still necessary to further explore their impact on the retrieval outcomes.

**Author Contributions:** M.Z.: conceptualization, investigation, data curation, methodology, writing—original draft preparation and editing, and software. M.W.: supervision, project administration, and funding acquisition, writing—review and editing. D.W.: supervision and project administration. S.W.: validation and formal analysis. W.X.: validation and visualization. All authors have read and agreed to the published version of the manuscript.

**Funding:** This research was funded by the National Natural Science Foundation of China (grant number 42272346), the Sichuan Science and Technology Program (grant number 2021YFG0319), the Opening Fund of Geomathematics Key Laboratory of Sichuan Province (grant number scsxdz2020yb08), and China Geological Survey (CGS) (grant number SYZXW2017101).

**Data Availability Statement:** Not applicable.

**Acknowledgments:** The authors are grateful for the helpful comments from the editors and reviewers, also the researchers and colleagues.

**Conflicts of Interest:** The authors declare that they have no known competing financial interest or personal relationships that could have appeared to influence the work reported in this paper.

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
