# Peer review of "Organic Matter Retrieval in Black Soil Based on Oblique Extremum Signatures"

_remotesensing, doi:10.3390/rs15102508_

Round 1

Reviewer 1 Report

This study aimed to provide a new insight into the indicative signatures of the spectral data as the oblique extremum points, not the traditional local extremum points also known as the absorption points.

However, as far as I know, this method based on the second derivative of soil reflection spectral curve extreme point is not innovative.

Other issues:

1 The authors do not provide an overview of the study area, and Figure 1 does not label the specific name of the study area. Moreover, The text marks in the figure are too small, resulting in blurring.

2 The authors did not elaborate on the soil sample treatment process, such as how to dispose of dead grass and other impurities in the soil. Moreover, the author did not introduce how to deal with the water in soil samples, so the influence of soil water on soil reflection spectrum was not eliminated. The effects of soil texture and aggregate structure on soil reflection spectra were not considered, or were not explained in the process of soil sample processing.

3 Why do you use K-means method to classify soil reflection spectra? Why are they divided into 4 categories?

4 Why use radial basis neural network algorithm, its advantage is? Why don't you do a comparison with other algorithms? In addition, complex neural network algorithm is not conducive to algorithm generalization.

I have no idea of the Quality of English Language 

Reviewer 2 Report

In general a nice paper with interesting content.

Line 19: if you mention "other two methods" in the Abstract you have to specify them.

Line 81: Table 2 appears in Chapter 3. It would be much better if you would move the table to Chapter 2.2. You may even merge Chapt. 3.1 with 2.2 , as it describes the spectral data and not the results, which start with Chapt. 3.2

Figure 1: Legend is too small. Please enlarge!

Figure 7: Legend is too small. Please enlarge!

Line 366 ff.: Please avoid to talk about "some types" and "some special transformations". Please be more concrete and specify types and transformations according to the nomenclature used in the paper

English needs to be improved to be better understandable.

Reviewer 3 Report

Dear Authors

The presented manuscript is written very well and is a solid work. May be published as is.

Best regards

Round 2

Reviewer 1 Report

Although some problems are difficult to further solve, such as the limitations of the generalized application of neural network algorithms, the author has revised most of my comments and the paper has met the requirements for publication.